# Transcriptome Analysis Reveals the Early Development in Subcutaneous Adipose Tissue of Laiwu Piglets

**DOI:** 10.3390/ani14202955

**Published:** 2024-10-14

**Authors:** Liwen Bian, Zhaoyang Di, Mengya Xu, Yuhan Tao, Fangyuan Yu, Qingyan Jiang, Yulong Yin, Lin Zhang

**Affiliations:** 1State Key Laboratory of Swine and Poultry Breeding Industry, National Engineering Research Center for Breeding Swine Industry, Guangdong Laboratory of Lingnan Modern Agriculture, Guangdong Provincial Key Laboratory of Animal Nutrition Control, College of Animal Science, South China Agricultural University, Guangzhou 510642, China; bianliwen2022@163.com (L.B.); dizhaoyang0410@163.com (Z.D.); 13955886100@163.com (M.X.); 18638611367@163.com (Y.T.); y18839502375@163.com (F.Y.); qyjiang@scau.edu.cn (Q.J.); 2Key Laboratory of Agro-Ecological Processes in Subtropical Region, Laboratory of Animal Nutritional Physiology and Metabolic Process, Institute of Subtropical Agriculture, Chinese Academy of Sciences, Changsha 410125, China

**Keywords:** Laiwu piglet, subcutaneous adipose tissue, transcriptome, carbohydrate and lipid metabolism, transcription regulation, proteostasis

## Abstract

**Simple Summary:**

Adipose tissue has A vital impact on animal production efficiency. Postnatal development plays a key role in regulating adipose tissue function; however, the underlying molecular mechanisms of this early-life programming of adipose tissue remain unclear. In this study, we analyzed the transcriptomes of adipose tissue between newborn and twenty-one days old Laiwu piglets, which is a Chinese indigenous breed characterized with high fat content. Our findings revealed a significant change in metabolic pattern as well as transcriptional and translational control in adipose tissue during this postnatal period, providing a better understanding of the development and function regulation of adipose tissue, improving pig production efficiency.

**Abstract:**

Adipose tissue plays an important role in pig production efficiency. Studies have shown that postnatal development has a vital impact on adipose tissue; however, the mechanisms behind pig adipose tissue early-life programming remain unknown. In this study, we analyzed the transcriptomes of the subcutaneous adipose tissue (SAT) of 1-day and 21-day old Laiwu piglets. The results showed that the SAT of Laiwu piglets significantly increased from 1-day to 21-day, and transcriptome analysis showed that there were 2352 and 2596 differentially expressed genes (DEGs) between 1-day and 21-day SAT in male and female piglets, respectively. Expression of genes in glycolysis, gluconeogenesis, and glycogen metabolism such as pyruvate kinase M1/2 (*PKM*), phosphoenolpyruvate carboxy kinase 1 (*PCK1*) and amylo-alpha-1, 6-glucosidase, 4-alpha-glucanotransferase (*AGL*) were significantly different between 1-day and 21-day SAT. Genes in lipid uptake, synthesis and lipolysis such as lipase E (*LIPE*), acetyl-CoA carboxylase alpha (*ACACA*), Stearoyl-CoA desaturase (*SCD*), and 3-hydroxy-3-methylglutaryl-CoA synthase 1 (*HMGCS1*) were also differentially expressed. Functional analysis showed enrichment of DEGs in transcriptional regulation, protein metabolism and cellular signal transduction. The protein–protein interaction (PPI) networks of these DEGs were analyzed and potential hub genes in these pathways were identified, such as transcriptional factors forkhead box O4 (*FOXO4*), CCAAT enhancer binding protein beta (*CEBPB*) and CCAAT enhancer binding protein delta (*CEBPD*), signal kinases BUB1 mitotic checkpoint serine/threonine kinase (*BUB1*) and cyclin-dependent kinase 1 (*CDK1*), and proteostasis-related factors ubiquitin conjugating enzyme E2 C (*UBE2C*) and cathepsin D (*CTSD*). Moreover, we further analyzed the transcriptomes of SAT between genders and the results showed that there were 54 and 72 DEGs in 1-day and 21-day old SAT, respectively. Genes such as *KDM5D* and *KDM6C* showed gender-specific expression in 1-day and 21-day SAT. These results showed the significant changes in SAT between 1-day and 21-day in male and female Laiwu pigs, which would provide information to comprehensively understand the programming of adipose tissue early development and to regulate adipose tissue function.

## 1. Introduction

Adipose tissue is an essential regulator of animal energy homeostasis and has a significant impact on pig production efficiency [1]. The primary role of adipose tissue is to store energy as fat in white adipocytes, occurring as hypertrophy (increase in size) and hyperplasia (increase in cell number), which directly affects animal fat content that has a vital impact on pig production efficiency [2]. Meanwhile, adipose tissue dynamically releases energy to meet animal requirements and serves as an essential regulator of glucose and lipid metabolism that could affect animal growth and survival, which are also vital factors of pig production efficiency [3]. In order to effectively regulate adipose tissue and improve pig production, a comprehensive understanding of adipose tissue development and function regulation is needed.

Studies have revealed that early life development and function of adipose tissue plays an important role in its physiology and affects future metabolic health [4,5]. The development of adipose tissue starts in utero, after that studies in rodents show that adipose tissue development is active during the early postnatal period [6]. Several human studies have also associated early life adipose tissue function and weight growth with later obesity [7,8]. These results suggest that early life is a crucial window of metabolic programming in adipose tissue and provides a critical opportunity for interventions on future adipose tissue function. However, the molecular mechanisms behind this period’s regulation of adipose tissue development and function remain unclear.

The Chinese indigenous breed, the Laiwu pig, provides an excellent model for studying adipose tissue since it has a high capacity for fat accumulation. The whole-body fat content of Laiwu pigs can reach 40%, and most of it accumulates in adipose tissue. Laiwu pigs have been applied to study the regulation of adipose tissue between different genetic backgrounds and nutritional factors [9,10]. However, most of these studies have been focused on growing pigs, and the regulation of adipose tissue during early life remains unknown.

Here, we aim to study the adipose tissue of 1- and 21-day old Laiwu piglets. By investigating the transcriptome of 1-day and 21-day adipose tissue, our results could provide insights into the programming mechanisms of adipose tissue early-life development and offer new perspectives to regulate adipose tissue function and improve pig production efficiency.

## 2. Materials and Methods

### 2.1. Animals

In total, 12 Laiwu piglets that were 1 day old and 12 Laiwu piglets that were 21 days old were sampled, each with 6 males and 6 females. All piglets were nursed by sows and fasted overnight before slaughter for tissue sample collection at the farm. Body weight was measured before sacrificing. Backfat thickness was measured at the first rib, last rib and the last lumbar vertebra locations using the vernier caliper [11]. All experiments were conducted in accordance with the Guide for the Care and Use of Laboratory Animals of South China Agricultural University (Permit Number 2022f125).

### 2.2. Tissue Sample Collection and RNA Extraction

The subcutaneous adipose tissue (SAT) samples were collected at the first rib location. Tissue samples were immediately placed in liquid nitrogen for snap-frozen and subsequently stored at −80 °C. Total RNA was extracted using TRIzol Reagent (Invitrogen, Carlsbad, CA, USA).

### 2.3. RNA-Seq Analysis

The RNA was processed to synthesize cDNA and prepare the sequencing library by the NovaSeq platform. Raw sequencing reads were quality-controlled by fastp, and an average of approximately 69M of high-quality clean reads were obtained. Then, sequencing reads were aligned to the reference genome (Sus scrofa 11.1) using HISAT2, and the alignment rates ranged from 95.28% to 96.7%. The average percentage of uniquely mapped reads and multiply mapped reads across all samples was 95.97% and 3.14%. Transcripts were assembled by StringTie. RSEM was used to quantify gene expression which was normalized by transcripts per million (TPM) values. Genes with *p*-value < 0.05 and |fold-change (FC)| ≥ 2 were designated as differentially expression genes (DEGs). The VennDiagram package (v3.5.0) was used to identify the overlapping genes.

### 2.4. Enrichment Analysis

Cluster of orthologous groups (COG) functional classification analysis was conducted by aligning transcriptome sequences with the EggNOG database. Functional enrichment analysis gene ontology (GO) and the examination of the Kyoto encyclopedia of genes and genomes (KEGG) were conducted by Goatools and KOBAS [12]. Principal components analysis (PCA) was performed using the scikit-learn library in Python. Volcano plots for DEGs distribution was visualized using R packages (v3.5.0).

### 2.5. PPI Network and Module Analysis

Protein–protein interaction (PPI) analysis was investigated using the STRING-db server (v12.0) with an interaction score threshold set at 0.4. Subsequently, hub genes of the networks were identified based on the Maximal clique centrality (MCC) method using the CytoHubba plugin (v0.1) in Cytoscape software (v3.10.0). The network visualization was achieved through Gephi software (v0.10.1).

## 3. Results

### 3.1. Body Weight and Backfat Thickness of 1-Day and 21-Day-Old Laiwu Piglets

From 1-day to 21-days, the body weight of male Laiwu piglets increased from 1.083 ± 0.04577 kg to 3.746 ± 0.3929 kg, and in female piglets the body weight increased from 1.082 ± 0.02866 kg to 3.896 ± 0.33254 kg (Figure 1A). We measured the backfat thickness at the first rib, last rib, and the last lumbar vertebra, and the results showed a significant increase at all three locations (Figure 1B–D), indicating the subcutaneous adipose tissue (SAT) significantly developed during this period. Meanwhile, with the similar body weight, female piglets of 21-days tended to have higher backfat thickness than males (*p* = 0.0775, *p* = 0.1659, *p* = 0.5603) (Figure 1B–D).

### 3.2. Transcriptome of Subcutaneous Adipose Tissues in Male Piglets

We analyzed the transcriptomes of SAT between 1-d and 21-d. In male piglets, solo-PCA analysis revealed a clear separation between 1-d and 21-d old piglets, indicating a significant difference in SAT transcriptomes (Appendix A). Differential gene expression analysis showed 2352 differentially expression genes (DEGs), among them 1067 were up-regulated while 1285 were down-regulated (Figure 2A). Functional analysis showed that these DEGs were enriched in carbohydrate and lipid metabolism, transcriptional regulation, protein metabolism, and cellular signal transduction (Figure 2B,C and Appendix A).

Glycolysis enzyme pyruvate kinase M1/2 (PKM), phosphofructokinase polypeptide X (PFKM), glyceraldehyde-3-phosphate dehydrogenase (GAPDH), phosphoglycerate mutase 2 (PGAM2), enolase 3 (ENO3), and lactate dehydrogenase (LDHA) showed lower expression in 21-d male SAT than 1-d, which could indicate that glycolysis might decrease from 1-d to 21-d in SAT (Table 1). However, PFKFB3, a glycolysis activator, was higher in 21-d SAT. Phosphoenolpyruvate carboxykinase 1 (PCK1), a key gluconeogenic enzyme [13], was higher in 21-d SAT. Phosphoglycolate phosphatase (PGP) was lower in 21-d SAT. AGL and PYGM, enzymes involved in glycogen degradation, were lower in SAT of 21-d. GBE1, glycogen branching enzyme, was also lower in 21-d SAT. These results showed that there were significant differences in carbohydrate metabolism between 1-d and 21-d SAT.

Lipolytic enzyme LIPE (HSL), was significantly higher in 21-d SAT (Table 1) (*p* < 0.01). PNPLA2 also known as anti-adipose triglyceride lipase (ATGL) was also higher in 21-d (*p* < 0.01), which catalyzes the first step of TG hydrolysis [14]. Glycerol-3-phosphate acyltransferase GPAT4 was higher in 21-d SAT, and GPAT3 was lower in 21-d SAT, which are TG synthesis enzymes [15]. Lipogenic enzyme ACACA was lower in 21-d SAT. Acetyl-CoA synthetases ACSL1 and ACSS2 were all lower in 21-d SAT, which are enzymes catalyzing long chain fatty acid CoA and short chain fatty acid CoA synthesis, respectively. Stearoyl-CoA desaturase (SCD), a key enzyme in lipid metabolism which synthesizes monounsaturated fatty acids, was lower in 21-d. ATP citrate lyase (ACLY), an enzyme that generates acetyl-CoA from citrate for fatty acid biosynthesis, was lower in 21-d. Fatty acid desaturation enzymes FADS1 and FADS2 were lower in 21-d SAT, suggesting the unsaturation of fatty acids could also be lower. Moreover, enzymes involved in sterol synthesis, MVK, IDI1, HMGCS1, and HMGCS2, were lower in 21-d SAT.

The mitochondrial inner membrane gene NDUFS1, component of complex I, was lower in 21-d SAT than 1-d, along with mitochondrial genes UQCRFS1 and UQCRQ, a component of complex III (Appendix A). Mitochondrial ATP synthase-encoding genes ATP5F1A and ATP5MC1 were lower in 21-d SAT. ND5, a mitochondrial gene encoding NADH dehydrogenase, was lower in 21-d SAT. NNT, an enzyme-consuming proton gradient to produce NADPH, was lower in 21-d. Malate dehydrogenase 2 (MDH2), an enzyme that catalyzes the reversible oxidation of malate to oxaloacetate, was down in 21-d. PDHB, DLAT, and PDHX, components of pyruvate dehydrogenase complex (PDC) were lower in 21-d SAT.

Functional analysis showed DEGs were enriched in the transcription regulation (Figure 3A). CEBPB, a member of the CCAAT/enhancer-binding protein (C/EBP) gene family, was higher in male 21-d SAT. FOXO4, a member of the forkhead box O family, was higher in 21-d SAT. In addition to FOXO4, other FOXO signaling factors including SGK2 and SOD2 were also higher in 21-d SAT. Activating protein 1 (AP-1) transcription factors, including FOS, JUNB, and ATF3, were significantly higher in 21-d SAT than 1-d (Figure 3A). The E2F1 transcription factor involved in the cell cycle was lower in 21-d SAT.

Perilipins PLIN1 and PLIN4, which are adipose-specific genes that encode lipid droplet-coating proteins, were significantly increased from 1-d to 21-d (Figure 3B). Ubiquitin-conjugating enzyme 2C (UBE2C) was lower in 21-d SAT, which directs ubiquitination to protein substrates for targeted degradation. Ubiquitination of proteins plays a vital role in protein metabolism, and in addition to UBE2C, the results showed factors in the protein ubiquitination system such as SKP2 and WWP2 were also significantly changed. Meanwhile, proteases CTSD, CTSF, CTSS, and CTSW were higher in 21-d SAT. Other factors involved in proteostasis such as PSMB9 and PSMF1 were also differently expressed between 1-d and 21-d in SAT.

The results showed that DEGs between 1-d and 21-d SAT were also enriched in signaling transduction (Appendix A). PPI network analysis of these DEGs showed that BUB1, CDK1, PBK, MELK, PLK4, and PLK1 were identified as potential hub genes. PIK3R3, a factor in the PI3K-Akt signaling pathway was higher in 21-d SAT. Adenylate cyclase 4 (ADCY4) and cAMP-responsive element binding protein 5 (CREB5), factors of the cAMP signaling pathway, were also different in 21-d SAT.

### 3.3. Transcriptome of Subcutaneous Adipose Tissues in Female Piglets

In female piglets, solo-PCA analysis also revealed a clear separation between 1-d and 21-d piglets, indicating a significant difference in SAT transcriptomes (Appendix A). There were 2596 DEGs identified, and 1273 were up-regulated, while 1323 were down-regulated (Figure 4A). Functional analysis showed these DEGs were enriched in carbohydrate and lipid metabolism, transcriptional regulation, protein metabolism, and cellular signal transduction (Figure 4B,C and Appendix A).

Glycolysis enzymes such as PKM and LDHA, glycolysis activator PFKFB3, gluconeogenesis enzymes PCK1 and PGP, and glycogen metabolic enzymes AGL and GBE1 were also significantly different in female SAT between 1-d and 21-d (Table 1). Meanwhile, key glycolysis enzyme hexokinase 2 (HK2) was significantly lower in 21-d female SAT. Moreover, in female piglets, glucose-6-phosphate dehydrogenase (G6PD), the rate-limiting enzyme in the pentose phosphate pathway (PPP), was significantly lower in 21-d SAT.

In lipid metabolism, female SAT shared common DEGs with male such as LIPE, ATGL, GPAT4, GPAT3, ACACA, ACSL1, ACSS2, ACLY, SCD, and FADS1, indicating the lipid uptake, TG synthesis, and lipolysis, fatty acid synthesis had similar changes to the males (Table 1). Specific to females, short chain fatty acid CoA synthesis enzyme ACSS3 was more significantly lower. The rate-limiting enzyme in cholesterol synthesis HMGCR was significantly lower in 21-d female SAT (Table 1), suggesting that the cholesterol synthesis might be even lower in 21-d female SAT. Meanwhile, HMGCS1 and HMGCS2 expression was lower in female 21-d SAT. MVD was lower, and more significant in females. These results suggest that cholesterol synthesis could be changing during the Laiwu piglet suckling period, and females were more severe than males.

Mitochondrial function was also different in females between 1-d and 21-d (Appendix A). In addition to NDUFS1, mitochondrial inner membrane genes NDUFA5, NDUFB3, and ATP5MC3 were significantly lower in females than males. In addition to ND5, mitochondrial genes ND1, ND2, ND3, ND4, and ND6 were significantly lower in female SAT. These results suggest that Laiwu female 21-d piglet subcutaneous adipose tissue might have even lower mitochondria function than males.

Transcription regulation was also significantly different in female SAT (Figure 5A). The results showed that DEGs also enriched in the FOXO signaling pathway in female SAT. In addition to FOXO4, FOXO1 and FOXO3 were also significantly higher in 21-d female SAT. These three FOXO transcription factors showed similar expression levels in newborn Laiwu SAT and were all upregulated in 21-d. Meanwhile, FOXO signaling-related factors, including IRS2 and CREBBP, were all expressed higher in 21-d than 1-d in female piglets. Instead of CEBPB in male SAT, another member of the C/EBP gene family CEBPD was significantly higher in female 21-d SAT. And, like males, activating protein 1 (AP-1) transcription factors, FOS and JUNB were significantly higher in 21-d than 1-d.

Like males, analysis of female SAT DEGs also showed enrichment in protein metabolism (Figure 5B). Adipose-specific PLIN1, PLIN4, and PLIN5 were significantly higher in 21-d female SAT than 1-d. UBE2C and CTSD, proteostasis-related factors were differentially expressed in female SAT. PPI network analysis showed that several heat shock proteins, including HSPB1, HSPD1, HSPE1, HSPB8, HSP90B1, HSPA4, HSPA6, and HSPA1L, were different between 1-d and 21-d female SAT. Among these, only HSPA6 and HSPA1L were higher in 21-d, others were all lower in 21-d. It is worth noting HSPA6 and HSPA1L expression levels were much lower than others.

Signal kinases BUB1, CDK1, PBK, MELK, and PLK4, related to the cell cycle, were also significantly different between 1-d and 21-d in females (Appendix A). PIK3R3 and ADCY4 were also higher in 21-d female SAT. In addition to these, the MAPK pathway was different. Mitogen-activated protein kinase 1 (MAP4K1) was significantly higher in 21-d female SAT, and MAP2K6 was lower. The upstream and downstream effectors such as RASGRF2, RASGRP1, RASGRP2, DUSP1, DUSP16, and PTPN7 were higher in 21-d SAT; CACNA2D1 and CACNG4 were lower in 21-d SAT.

### 3.4. Comparative Analysis of Male and Female Subcutaneous Adipose Tissues Transcriptomes

Adipose tissue showed gender differences in its regulation and function [16,17], and our results also showed that 21-d old female Laiwu piglets tended to have higher SAT than maleas. Transcriptome analysis of 1-d and 21-d SAT in male and female piglets showed that there were 1388 common genes with 1208 female-specific genes and 964 male-specific genes (Appendix A). These results suggest that the changes in transcriptomes in SAT from 1-d to 21-d presented both similarity and specificity between males and females. Therefore, we further analyzed the transcriptomes between genders. The results showed that there were 54 DEGs between males and females in 1-d Laiwu piglet SATs, among them, 18 were higher in females and 36 were higher in males. Meanwhile, in 21-d Laiwu piglets, there were 72 DEGs and 15 were higher in female and 57 were higher in male (Figure 6A).

The results showed that the histone demethylation enzymes KDM5D and KDM6C were expressed in male piglet SAT, however, in female the expression was undetected in both 1-d and 21-d (Figure 6B). The results also found that EIF1A, EIF2S3Y, and DDX3Y showed high expression in male SAT, but little to no expression in female in both 1-d and 21-d Laiwu piglets (Figure 6C). USP24 was also differentially expressed between genders with significantly higher expression in males in both 1-d and 21-d SAT.

There were DEGs specific in 1-d SAT (Figure 6D). Transcription factor CEBPD was higher in male than female in 1-d piglet SAT. TXNIP, thioredoxin binding protein, was also higher in male in 1-d SAT. HSPA5, a member of the heat shock protein 70 family and a regulator of endoplasmic reticulum (ER) homeostasis, was higher in female piglet SAT in 1-d piglets.

There were also DEGs specific in 21-d SAT (Figure 6E). DKK3 showed significantly higher expression in female than male in 21-d piglet SAT. The expression of DKK3 in 1-d SAT showed no difference between genders, and from 1-d to 21-d, DKK3 expression in males tended to decrease and in females, it tended to increase.

## 4. Discussion

Adipose tissue has important roles in animal health and affects production efficiency. Studies have shown that early life is a vital period for adipose tissue development and function in humans and mice [4]; however, the understanding of pig adipose tissue in this period is very limited. Here, we studied the adipose tissue in 1-d and 21-d piglets of the Laiwu breed, which is characterized by high adipose tissue content. The results showed that during this period, Laiwu piglet adipose tissue significantly increased. The backfat thickness of the first rib location in 1-d piglets was 1.436 ± 0.129 mm, and in 21-d piglets, the backfat thickness reached 9.6886 ± 0.663 mm. These results indicated that the adipose tissue of Laiwu piglets had significant development from newborn to 21 days, and understanding of the mechanisms behind the regulation of this period could provide a vital opportunity for regulation of adipose tissue accumulation.

The transcriptomes analysis of adipose tissue showed that there were significant differences in the metabolic pattern between 1-d and 21-d. First, carbohydrate metabolism had changed. Glycolysis enzymes including *PKM*, *GAPDH*, *PGAM2*, *ENO3*, *LDHA,* and *HK2* were expressed lower in 21-d SAT than 1-d which might suggest a lower glycolysis level. However, there was also contrary evidence such as glycolysis activator *PFKFB3* was higher in 21-d SAT. Studies have shown that *PFKFB3* linked glycolysis with cell proliferation, especially in cancer cells [18,19], yet the role of *PFKFB3* in pig adipose tissue remains unknown. The rate-limiting enzyme of the PPP pathway G6PD [20] was differentially expressed between 1-d and 21-d in female Laiwu SAT, suggesting the PPP pathway could also be changing in SAT during this period. As the initial step and the basis of the metabolic network, glycolysis is a very important and complex process. Here our results suggested that glycolysis was significantly changed between 1-d and 21-d, however, the exact effects need further validation.

Meanwhile, the higher expression of gluconeogenic enzymes such as *PCK1* [13] and lower expression of *PGP* which studies showed could negatively regulate gluconeogenesis [21], suggested that gluconeogenesis might be higher in 21-d Laiwu pig SAT. Of note, the results showed that pyruvate carboxylase (PC), which catalyzes the carboxylation of pyruvate to oxaloacetate, was higher in 21-d SAT. The detected expression level of PC was high in Laiwu SAT, and previous studies showed that PC mainly regulated fatty acid synthesis in adipose tissue, and regulated gluconeogenesis in the liver [22,23]. However, our results showed no clear indication that the lipogenic capacity was higher in 21-d SAT, and whether the higher PC expression in 21-d SAT contributes to a higher gluconeogenesis requires further investigation.

Moreover, glycogen metabolic enzymes such as *AGL* and *GBE1* showed differential expression. Glycogen metabolism is considered primarily in the liver and muscle; however, recent studies have suggested that glycogen metabolism, although very low, could serve as an energy sensing mechanism in adipose tissue [24]. Our results showed that glycogen metabolism might be different from 1-d to 21-d in pig SAT, and the exact role of this change in adipose tissue function needs further investigation.

Lipid metabolism was also changed from 1-d to 21-d in Laiwu pig SAT. LIPE and ATGL were significantly higher in 21-d SAT than 1-d, and TG synthesis enzyme GPAT4 was also higher in 21-d SAT [15]. Even though GPAT3 was lower in 21-d SAT, our results showed that GPAT4 had a significantly higher expression level than GPAT3 in Laiwu piglet SAT. These results suggest that lipid uptake, TG synthesis, and hydrolysis might be higher in 21-d SAT. DEGs such as ACACA, ACSL1, ACSS2, ACLY, and SCD suggested that lipogenesis and fatty acid synthesis might be lower. These results might suggest that lipid absorption could be a major contributor to increased adipogenesis, and lipid synthesis might not be significant in 21-d SAT; other organs and tissues such as the liver, or even the intestines, could serve as the main sites for lipogenesis. Moreover, TG turnover was upregulated from 1-d to 21-d, together with higher gluconeogenesis in 21-d SAT, which would provide sufficient energy to support the fast growth during this period.

Amino acid metabolic enzyme GOT1 was significantly lower in female 21-d SAT (Appendix A). Studies have shown that GOT1 regulates cell proliferation by participating in amino acid, especially glutamine, metabolism [19]. There was also a study that showed that GOT1 might be involved in porcine adipocyte differentiation [20]. Here, the results showed that GOT1 was less expressed from newborn to 21-d, suggesting it might participate in adipose differentiation and development, yet the mechanisms need further validation.

Mitochondrial inner membrane genes such as *NDUFS1* [25], and NDs, mitochondrial genes encoding NADH dehydrogenase and function in mitochondrial electron transport [26], were significantly lower in 21-d than 1-d adipose tissue. MDH2, an enzyme involved in the TCA cycle [27], and PDHB, DLAT, and PDHX, components of the pyruvate dehydrogenase complex (PDC) which converses pyruvate to acetyl CoA in mitochondria and links glycolysis and TCA cycle [28], were all lower in 21-d SAT. These results might suggest a lower level of energy production through glycolysis and TCA cycle in 21-d SAT. Our results showed that the mitochondrial function of Laiwu SAT could be lower in 21-d than 1-d, which indicated that the energy production inside the adipose tissue could be low, which would contribute to high adipogenesis.

Cell proliferation and differentiation are important factors that regulate tissue development. Our results showed that many genes involved in cell cycle control were differentially expressed in Laiwu piglet SAT from 1-d to 21-d, including transcription factors such as *FOXOs*, *AP-1* [29], and *E2F1* [30]; DNA replication enzymes and factors such as *DNA2* and *MCMs*; and signaling molecules such as *BUB1* and *CDK1* [31]; and factors in the p53 signaling pathway *GADD45G* and *GADD45B*. The FOXO transcription factor family, first identified as a mediator of insulin action, is known to regulate multiple processes including glucose and lipid metabolism, energy production and cell growth and differentiation. *FOXO* signaling interacts with many factors and conditions [32]. Our results showed that FOXO4, FOXO3, and FOXO1 as well as FOXO signaling-related factors SGK2, SOD2, IRS2, and CREBBP [33,34,35] were all differentially expressed between 1-d and 21-d SAT. This suggests that the *FOXO* pathway is important during adipose tissue development, therefore, the mechanisms behind the *FOXO* signaling regulation in pig adipose tissue need further investigation. There were studies that showed that *FOS*, a member of AP-1, might be involved in pig adipose development in the gestational period [36]. Even though E2F1, an important transcriptional factor involved in the cell cycle, was lower in 21-d SAT than 1-d, there were recent studies that showed that, in addition to cell cycle regulation, *E2F1* had a pro-apoptosis effect and could affect metabolism [37,38]. The exact roles of these genes in adipose tissue hyperplasia and the connections between them are worth further investigation and would provide potential targets for effective modulation of adipose tissue development as well as function. 

Cell differentiation would be another factor that participates in newborn piglet adipose tissue development. The results showed that *BMP6*, a known regulator of adipose tissue development, was higher in 21-d SAT. BMPs are part of the superfamily of transforming growth factor-beta (*TGF-β*) ligands and studies showed that BMP6/TGF-β signaling may regulate adipocyte differentiation [39]. These results could indicate that cell proliferation and differentiation were all changed in piglet adipose tissue from newborn to 21-d. In addition to these known factors involved in cell proliferation and differentiation, the results also showed genes such as *ACOT7*, an enzyme that catalyzes the hydrolysis of fatty acyl-CoAs to free fatty acids and CoA-SH, were significantly lower in 21-d SAT. There were that studies showed that *ACOT7* might be involved in the cell cycle [40], therefore, DEGs such as *ACOT7* might be involved in porcine adipose development during the suckling period, and further studies are needed for validation.

Inflammation plays a vital role in adipose tissue function since there are a significant number of immune cells residing in adipose tissue which can change in number and type to regulate adipocyte function [41,42]. Our results showed that there were immune-related DEGs, such as *SLA-DQB1*, *SLA-DOA*, *HLA-DRA*, *CD74,* and *B2M*, members of major histocompatibility complex (MHC) system [43,44], which were higher in 21-d SAT than 1-d. These genes are involved in immune cell antigen processing and presentation and the higher expression of these genes might indicate a high infiltration of immune cells into adipose tissue. The results also showed that the action of the complement system might be different. Complement factor D (*CFD*), was significantly higher in 21-d SAT. *CFD* is an essential factor in the activation of the alternative pathway (AP) and is produced mainly by adipose tissue [45,46]. Our results also showed that *MASP1* and *MASP2* were higher in 21-d, and there were studies showed that *MASP1* was involved in *CFD* activation [47]. In addition to *CFD*, complement system factors such as *C1S*, *SERPING1,* and *PLAT* were all higher in 21-d SAT in both genders. *C1R* and *C2* were significantly higher in males, and *C1QA* and *C1QB* were higher in females. These results could indicate that the inflammation of adipose tissue had changed from 1-d to 21-d. However, whether this change is a causal effect of adipose tissue development remains unknown. Further studies would be needed to elucidate the inflammation change in immune cells and adipocytes in detail to help better understand their role in pig adipose tissue development and function regulation.

Our results showed that circadian regulators *PER1* and *PER2* [48] were higher in 21-d female SAT, and *TIMELESS* was lower. Circadian rhythm has great impacts on adipose tissue, such as lipid metabolism, energy expenditure, inflammation, and adipokine secretion [49]. *PER1*, *PER2,* and *TIMELESS* are important circadian regulators; however, their roles in pig adipose tissue remain unclear and need further validation.

The results showed that the expression of certain heat shock proteins including *HSPB1*, *HSPD1*, *HSPE1*, *HSPB8*, *HSP90B1*, *HSPA4*, *HSPA6,* and *HSPA1L* were significantly different between 1-d and 21-d female SAT. These heat shock proteins are chaperones that facilitate protein folding and maintain proteostasis and can interact with different proteins to affect multiple cellular processes such as mitochondrial function and autophagy [50,51]; however, their function in adipose tissue remains unclear, which needs further investigation.

DEGs are related to retinoid regulation. *CRABP2*, retinoic acid (RA) and fatty acid binding protein, a shuttling protein which facilitates *RA* binding to its receptor complex, was lower in 21-d SAT. Retinol-binding protein *RBP5* was also lower. *CRABP1* expression levels were lower than *CRABP2* in 21-d SAT. Aldehyde dehydrogenases *ALDH1A1* and *ALDH1A2*, studies showed that they were involved in RA synthesis by catalyzing conversion of retinaldehyde to retinoic acid [52], were also differentially expressed in SAT between 1-d and 21-d. These results might suggest that vitamin A as a potential method to regulate pig adipose development and function in this period and the mechanisms need further investigation.

Studies have shown that adipose tissue has presented with gender-specific physiology; however, the mechanisms behind the gender differences remain unclear. Here, we found that female Laiwu piglets tended to accumulate more adipose tissue than males from 1-d to 21-d. Moreover, when we compared the transcriptomes between 1-d and 21-d SAT in male and female Laiwu piglets, the results showed that there were only approximately 40% DEGs that were overlapped between genders, and over 60% DEGs showed gender-specific differential expression from 1-d to 21-d development. These results suggest that there are both common and specific factors that affect adipose tissue development in the early life of male and female piglets.

Further analysis comparing transcriptomes of male and female SAT showed that there were genes with significant differential expression between genders. Genes such as *KDM5D* and *KDM6C* showed expression in male SAT, and were undetected in females in both 1-d and 21-d piglets, suggesting that they might have specific roles in male SAT development and function regulation. These genes belong to the histone demethylase family and modify histone methylation [53,54], and, as an important epigenetic regulatory mechanism, studies have shown that histone methylation plays an important role in adipose development and function [55]. The exact function of *KDM5D* and *KDM6C* in porcine adipose tissue remains unclear and requires further investigation. Our results showed that *EIF1A*, *EIF2S3Y*, *DDX3Y,* and *USP24* also had gene-specific expression in Laiwu SAT. Which are involved in transcription and translation regulation as well as ubiquitin-dependent protein catabolism. They might be gender-specific regulators and their regulation is worth further investigation. Further validation of these genes’ roles in adipose tissue could provide gender-specific targets for regulation.

Moreover, our results also showed genes such as ENSSSCG00000012179 and ENSSSCG00000053830 had differential expression between genders in SAT; however, their functions in pigs remain unknown. All these results showed the gender differences in pig adipose tissue and provided information for comprehensive understanding of the gender differences in adipose tissue. For future scientific research and pig production, gender differences in adipose tissue should be considered.

Adipose tissue function is vital for animal production. Previous studies have investigated the late stage of adipose tissue in Laiwu pigs, and these studies have compared the Laiwu adipose with lean-type pigs and their results showed that adipose tissue in Laiwu pigs had higher lipid synthesis capacity [10,56,57]. The early development of adipose tissue could have significant impacts on its function, therefore, understanding the mechanisms behind developmental regulation would be important. Our results showed that there were many differentially expressed genes between newborn and 21-d piglet SATs. These genes might be involved in carbohydrate and lipid metabolism, energy homeostasis, cell cycle, inflammation, etc., and have potential roles in adipose tissue development and function regulation. In addition to these perspectives, there could be more clues in understanding pig adipose tissue since our results also showed DEGs with unknown function in adipose tissue and we would continue our investigation in future. Meanwhile, these identified DEGs would also need further investigation to validate their exact role and mechanisms. Adipose tissue as a vital regulator in energy homeostasis affects not only animal production but also human health. Here, we analyzed the adipose tissue from newborn to 21 days in healthy piglets which can provide new perspectives to fully understand early life adipose tissue development and function regulation, and benefit not only animal production, but also human health.

## 5. Conclusions

The subcutaneous adipose tissue significantly increased from 1-d to 21-d in Laiwu piglets. Transcriptome analysis showed 2352 and 2596 differentially expressed genes between 1-day and 21-day SAT in male and female piglets, respectively, which were enriched in carbohydrate and lipid metabolism, transcriptional regulation, protein metabolism, and cellular signal transduction. Potential key genes in these pathways were identified, including FOXO4, CEBPD, BUB1, CDK1, UBE2C, and CTSD. These findings provide new insights into a comprehensive understanding of adipose tissue development and function regulation and thus benefit pig production.

## Figures and Tables

**Figure 1 animals-14-02955-f001:**
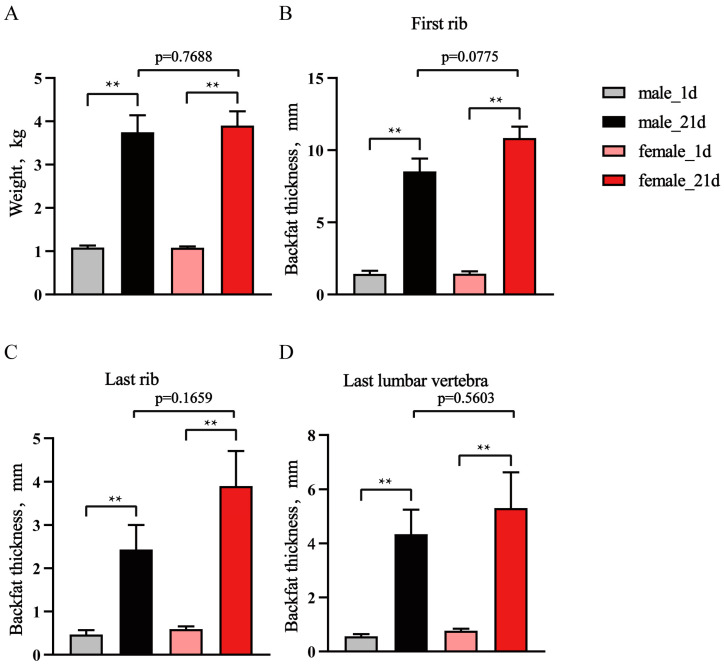
Weight and backfat thickness of Laiwu piglets. (**A**) Body weight of 1-d and 21-d old male and female piglets (n = 6). (**B**–**D**) Backfat thickness at the first rib (**B**), the last rib (**C**), and the last lumbar vertebra (**D**). ** *p* < 0.01.

**Figure 2 animals-14-02955-f002:**
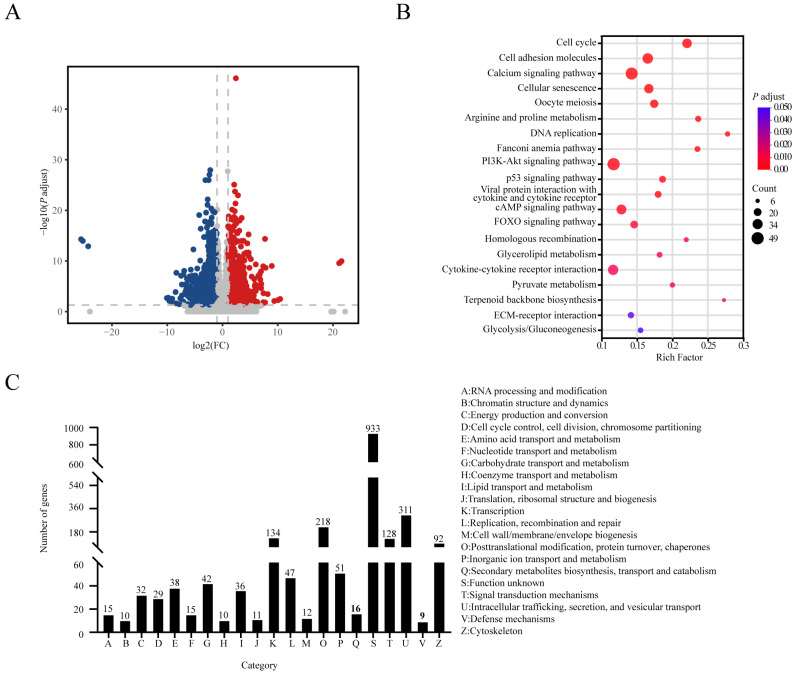
Transcriptome analysis of SAT in male Laiwu piglets. (**A**) Volcano plots of DEGs between 1-d and 21-d in SAT. The x-axis represents the fold change (FC) and the y-axis represents the *p*-value. Red dots indicate significantly up-regulated genes and blue dots indicate down-regulation. (**B**) KEGG pathways enrichment of DEG. (**C**) COG functional classification of DEGs between 1-d and 21-d in SAT.

**Figure 3 animals-14-02955-f003:**
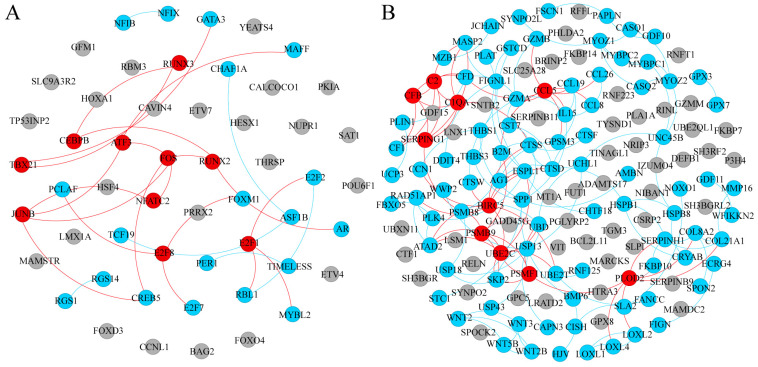
PPI network analysis of SAT in male Laiwu piglets. (**A**) PPI diagram of DEGs between 1-d and 21-d SAT in transcription category of COG classification. (**B**) PPI diagram of DEGs between 1-d and 21-d SAT in posttranslational modification, protein turnover, and chaperone category of COG classification. Red nodes indicate predicted hub genes, blue nodes indicate genes with more than one connection, and gray nodes indicate genes with no connection.

**Figure 4 animals-14-02955-f004:**
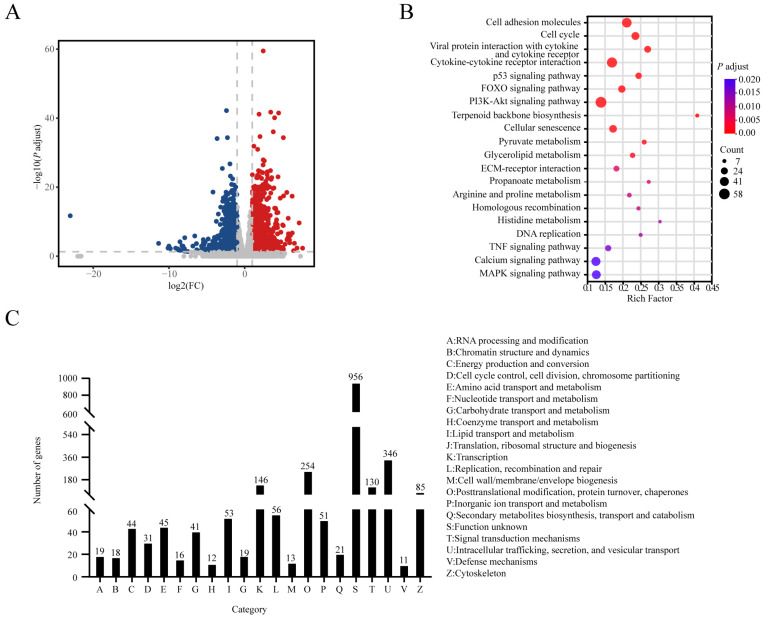
Transcriptome analysis of SAT in female Laiwu piglets. (**A**) Volcano plots of DEGs between 1-d and 21-d in SAT. (**B**) KEGG pathways enrichment of DEG. (**C**) COG functional classification of DEGs between 1-d and 21-d in SAT.

**Figure 5 animals-14-02955-f005:**
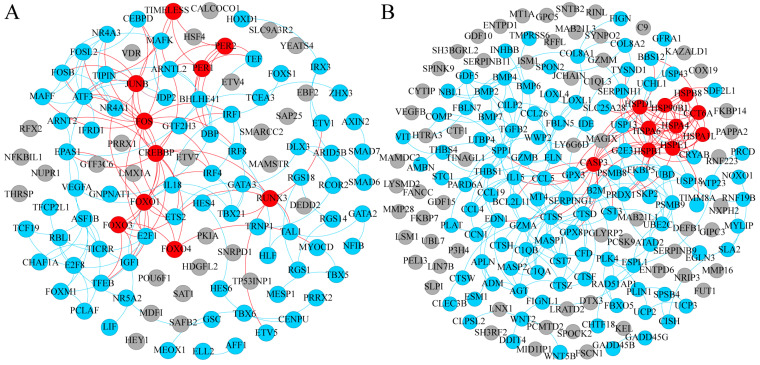
PPI network analysis of SAT in female Laiwu piglets. (**A**) PPI diagram of DEGs between 1-d and 21-d SAT in transcription category of COG classification. (**B**) PPI diagram of DEGs between 1-d and 21-d SAT in posttranslational modification, protein turnover, and chaperone categories of COG classification. Red nodes indicate predicted hub genes, blue nodes indicate genes with more than one connection, and gray nodes indicate genes with no connection.

**Figure 6 animals-14-02955-f006:**
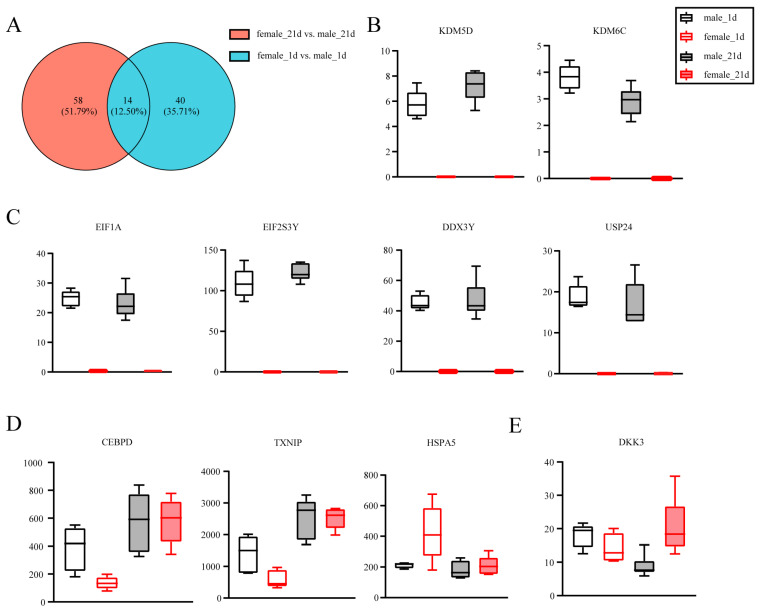
Transcriptome analysis of SAT in Laiwu piglets between genders. (**A**) Venn diagram of DEGs. (**B**–**E**) Expression levels of DEGs in SAT of 1-d and 21-d old Laiwu piglets.

**Table 1 animals-14-02955-t001:** DEGs in carbohydrate and lipid metabolism.

Gene ID	Gene Symbol	Male		Female	
		21 d vs. 1 d		21 d vs. 1 d	
		Log2(FC)	Padjust	Log2(FC)	Padjust
ENSSSCG00000001930	PKM	−1.396	3.90 × 10^−8^	−1.087	6.94 × 10^−4^
ENSSSCG00000021129	PFKM	−2.522	3.19 × 10^−7^	−1.434	1.05 × 10^−1^
ENSSSCG00000000694	GAPDH	−1.015	4.07 × 10^−3^	−0.582	2.51 × 10^−1^
ENSSSCG00000016720	PGAM2	−4.443	7.95 × 10^−7^	−1.572	3.88 × 10^−1^
ENSSSCG00000017904	ENO3	−4.286	5.77 × 10^−7^	−1.298	4.86 × 10^−1^
ENSSSCG00000013366	LDHA	−1.9	1.24 × 10^−10^	−1.577	4.06 × 10^−6^
ENSSSCG00000011133	PFKFB3	1.31	8.23 × 10^−7^	2.052	2.96 × 10^−15^
ENSSSCG00000007507	PCK1	1.335	2.26 × 10^−3^	1.348	4.97 × 10^−6^
ENSSSCG00000036614	PGP	−1.24	3.30 × 10^−4^	−1.307	6.87 × 10^−5^
ENSSSCG00000006872	AGL	−1.872	2.70 × 10^−10^	−1.636	2.55 × 10^−6^
ENSSSCG00000013022	PYGM	−4.107	3.88 × 10^−5^	−2.187	1.57 × 10^−1^
ENSSSCG00000012000	GBE1	−1.29	2.67 × 10^−5^	−1.113	4.62 × 10^−4^
ENSSSCG00000008261	HK2	−0.971	1.19 × 10^−1^	−1.49	2.98 × 10^−5^
ENSSSCG00000025108	G6PD	−0.888	1.87 × 10^−2^	−1.312	5.69 × 10^−6^
ENSSSCG00000003018	HSL	1.314	5.28 × 10^−7^	1.618	1.71 × 10^−16^
ENSSSCG00000012841	ATGL	1.238	5.11 × 10^−9^	1.237	2.12 × 10^−10^
ENSSSCG00000007019	GPAT4	1.429	5.46 × 10^−10^	1.582	1.78 × 10^−23^
ENSSSCG00000009233	GPAT3	−1.403	4.84 × 10^−7^	−1.285	1.48 × 10^−11^
ENSSSCG00000017694	ACACA	−1.836	8.31 × 10^−5^	−1.508	7.66 × 10^−4^
ENSSSCG00000017421	ACLY	−2.466	6.73 × 10^−8^	−2.903	1.17 × 10^−16^
ENSSSCG00000015784	ACSL1	−1.675	5.36 × 10^−6^	−2.539	1.48 × 10^−16^
ENSSSCG00000007286	ACSS2	−1.434	6.71 × 10^−4^	−1.437	9.68 × 10^−5^
ENSSSCG00000000939	ACSS3	−0.9	6.15 × 10^−3^	−1.067	1.71 × 10^−5^
ENSSSCG00000010554	SCD	−3.152	3.19 × 10^−3^	−2.206	1.88 × 10^−2^
ENSSSCG00000024015	FADS1	−1.015	1.31 × 10^−3^	−1.336	1.89 × 10^−5^
ENSSSCG00000013072	FADS2	−1.056	8.34 × 10^−8^	−0.99	5.61 × 10^−5^
ENSSSCG00000009931	MVK	−1.281	1.81 × 10^−3^	−1.387	5.17 × 10^−7^
ENSSSCG00000029066	IDI1	−1.591	1.67 × 10^−4^	−1.448	7.69 × 10^−5^
ENSSSCG00000014080	HMGCR	−0.636	4.34 × 10^−2^	−1.096	1.46 × 10^−5^
ENSSSCG00000016872	HMGCS1	−1.25	7.90 × 10^−5^	−1.756	1.43 × 10^−12^
ENSSSCG00000036808	HMGCS2	−2.849	3.41 × 10^−3^	−1.661	9.61 × 10^−3^

ID: identifier, FC: fold change.

## Data Availability

Data are available upon request to the corresponding author.

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
