# Peer review of "Transcriptome Analysis Reveals the Early Development in Subcutaneous Adipose Tissue of Laiwu Piglets"

_animals, 2024, doi:10.3390/ani14202955_

Round 1
Reviewer 1 Report
Comments and Suggestions for Authors
The subcutaneous fat on the back of pigs is a key indicator of the amount of fat deposited in the carcass, and studying its formation mechanism is of great significance for the improvement of the traits. The author selected the samples from day 1 and day 21 for comparison, which can enrich the data of the early occurrence of fat.
1. In the phenotypic results, the subcutaneous fat on day 1 was about 1mm, which was easily contaminated by the dermis during the sampling process. HE staining results should be provided to ensure the accuracy of the tissue morphological characteristics.
2. Phenotypic analysis showed that the amount of fat deposition on day 21 was much higher than that on day 1, but there was no pathway related to lipid generation in differential gene enrichment analysis. What is the reason?
3. The list of genes with significant differences showed that at 21 days, the gene HSL of lipysis was significantly up-regulated, while the genes SCD, FADS1 and FADS2 of lipgenesis were significantly down-regulated, which was contrary to the biological process of lipgenesis. Why? The authors should provide qPCR and protein quantitative results of related genes to determine their reliability.
4. The materials used in this study were samples of 1 and 21 days of birth, and a large number of pathways related to inflammation and oxidative stress were indeed enriched. How to explain this, it is suggested to conduct quantitative verification of differential genes.
5. The author mentioned in the preface that there have been many studies on the late stage of fat under pig skin, and the important biological processes in this study should be compared with the existing reports, showing the characteristics of continuous changes in the fat formation process in a longer period.
Reviewer 2 Report
Comments and Suggestions for Authors
Overall comments
The manuscript presents interesting results. Unfortunately there is some crucial information missing in the described design of the study. Results and discussion should be revised and rewritten. Unification of the names used for all groups of animals and clear indication of statistically significant changes should be made.
Introduction
Introduction is well structured, gives adequate background for the reader and explains the animals enrolled in the study. However, it needs some major changes. First paragraph requires adequate references. Last paragraph describes the results of the study, which should be reserved for the results and discussion. Please rewrite this section to reflect the aim of presented research/ hypothesis/value for the field, etc.
Material and methods
Line 88 – do you mean “studied animals”?; line 91: I do not believe sacrificing is a suitable or necessary word to use in a scientific paper; were the animals killed after second sampling?; line 93: should include the permit number;
This section should include the information about the conditions in which animals were kept (e.g. whether the tissue collection was performed at the farm or in the laboratory, etc.), adipose tissue sampling equipment, method of assessing the thickness of adipose tissue, anaesthesia protocols, the reagents kits used for RNA isolation, RNA quality check, cDNA synthesis.
Results
Line 131: lacks p-value
Line 132: lacks p-value which will clearly indicate that those differences were insignificant (according to the Figure 1)
Unnecessary references throughout this section. Information relating to the references should be provided in the discussion.
Line 166- provide p-value
Table 1 lacks marking the significant differences.
Results should be rewritten and shortened. This section should just describe the obtained results. In current form there is too much information belonging in the discussion. Figures 2 and 4 should be slightly bigger – they would be easier to read.
Discussion
This section provides an adequate overview of the obtained results, however, some mentioned points need to be reinforced by appropriate references. Moreover, this section should be rewritten and then evaluated by a native speaker to avoid grammatical mistakes and ensure proper formal and scientific choice of words.
Lines 462-464: since there is no information about the pigs environment and its changes or any possible oxidative stress causes between samplings it’s hard to draw the conclusion about different redox homeostasis in piglets of different age.
Conclusions
This paragraph should be slightly rewritten to strengthen obtained results in DEGs between animals.
Additional files
Figure S4 – please rewrite the legend to name the groups of animals clearly - e.g. female_21d instead of female_21. It reads better and is clearer to the detail-oriented readers.
Figure S5 is presented form is difficult to read. There is no information what red lines in the x axis mean (left side of the diagram) or if SD or SEM is presented in the expression levels.
Comments on the Quality of English LanguageArticle is understandable for the reader, however there are some mistakes. I suggest proofreading by a native speaker or the use of proofreading services.
Round 2
Reviewer 1 Report
Comments and Suggestions for Authors
The author basically states the problems of the premise clearly and agrees to publish.
Author Response
Comments 1: The author basically states the problems of the premise clearly and agrees to publish. Response 1: Thank you for the comment.